# Learning with Online Teaching Video Cases: Investigating Pre-Service Preschool Teachers' Perceived Usefulness and Needs

Rongrong Xu * , Alfredo Bautista and Weipeng Yang

Department of Early Childhood Education, Faculty of Education and Human Development,
The Education University of Hong Kong, Hong Kong; abautista@eduhk.hk (A.B.); wyang@eduhk.hk (W.Y.)
* Correspondence: s1142491@s.eduhk.hk

**Abstract:** Extensive empirical research has emphasized the benefits of integrating Online Teaching Video Cases (OTVCs) into pre-service preschool teacher education. However, there is a research gap concerning the perceptions and needs of pre-service preschool teachers regarding OTVCs. This cross-sectional study, therefore, investigated pre-service preschool teachers' perceptions of usefulness and need pertaining to OTVCs and examined potential differences across course year levels. A self-designed questionnaire survey was completed by 744 participants from the Shandong Province (China), with a focus on five domains: actors showcased in the OTVCs, OTVC-mediated activities, learning facilitators, situations requiring OTVCs, and areas for improvement. The results revealed that the participants identified expert teacher demonstrations and collaborative learning experiences with peers and instructors as the most useful types of OTVCs. They expressed the need for OTVCs to assist them with job preparation and a desire to learn content knowledge and engage with larger communities of preschool practitioners. Interestingly, the findings revealed significant differences among participants of different year levels, with Year 3 participants finding OTVCs more useful and necessary in most domains. These findings will help preschool teacher educators improve the responsiveness of OTVC-based instruction, thereby providing online video resources tailored to the preferences and needs of pre-service preschool teachers.

**Keywords:** online teaching video cases; pre-service preschool teachers; perceived usefulness; needs; year levels

## 1. Introduction

In this study, we adopted a self-developed quantitative survey to examine the perceived usefulness and needs of pre-service preschool teachers (PPTs) regarding online teaching video cases (OTVCs). Drawing on existing literature [1–3], we identified five key domains regarding which scholars have called for further research on how to use videos to aid the learning of PPTs. These domains include the actors showcased in the videos, the learning activities employed, the learning facilitators involved, the situations in which videos are needed, and finally, the improvements observed in both teachers and children. Furthermore, we explored whether the preferences and needs for OTVCs varied among PPTs at different course year levels within teacher education programs. The subsequent section provides a comprehensive review of the literature on perceptions of videos as a learning tool and differences based on course year levels. Additionally, we conducted a detailed review of the aforementioned five domains, which are of particular interest to researchers studying pre-service teachers' video-mediated learning.

## 2. Literature Review

### 2.1. Pre-Service Teachers' Perceptions of Video Adoption and Differences across Course Year Levels

Video cases are becoming more prevalent worldwide for assisting pre-service teachers in their pedagogical development and growth [1]. To ensure the responsiveness and relevance of video-mediated training activities, empirical evidence has emphasized the significance of investigating pre-service teachers' perspectives on the usefulness of and need for videos [4]. In reviewing the literature, some studies exploring pre-service teachers' perspectives on video adoption conducted with elementary teachers are illuminating. For instance, Yadav [5] investigated pre-service elementary teachers' perception of their learning with teaching video cases and indicated that they preferred scaffolded learning with videos to improve their literacy instruction and observational abilities. Similarly, Kurz, Batarelo and Middleton [6] studied the need for teaching video cases from the perspectives of pre-service elementary teachers. They found that pre-service teachers had an acute need for instructional guidance when using video cases, including the development of lesson plans, the inclusion of expert analysis, and the acquisition of classroom management and child interaction skills.

In the field of preschool education, with the rapid advancements in digital technology and evolving learning environments, videos have gained popularity as e-learning mediums within PPT education programs [7]. In this context, researchers have begun exploring the impact of videos on PPTs' learning. In the research conducted by Santagata and Guarino [8], classroom videos were found to have great potential to assist PPTs in analyzing, reflecting on, and developing their skills of noticing classroom instructions and children. In the same vein, the scoping review conducted by Bautista, Ho, and colleagues [2] revealed that PPTs obtained multiple benefits from learning with classroom videos, including instructional skills, content knowledge, classroom management skills, teacher beliefs, and teacher-child interaction skills. However, it is worth noting that there is a lack of research specifically examining the practices that pre-service teachers find useful and their specific needs regarding online videos, particularly in the field of preschool education.

Pre-service teachers encounter different challenges and demands at different stages of their learning journeys, and their beliefs undergo changes throughout their studies [9,10]. Thus, the variable course year level constitutes a potential factor influencing student teachers' thinking. Fadlelmula [9] investigated the attitudes toward the teaching profession among pre-service mathematics teachers and discovered that the pre-service teachers in higher year levels had fewer positive attitudes than those in lower years. Similarly, Ambrosetti [11] concluded that pre-service teachers' specific needs for mentoring were connected to the extent of their progress through their program. First-year student teachers paid more attention to learning the teaching process, while final-year student teachers were more likely to polish their teaching as they prepared to enter the teaching profession. These findings suggest that the course year level could be an important factor to consider when analyzing pre-service teachers' perceptions and needs.

This study concerns the application of OTVCs in preparing PPTs in mainland China. Preschool education programs in Chinese colleges typically span three years, each of which has different learning priorities [12]. During the first year, programs provide students with generic courses on child development and psychology. In the second year, the focus shifts towards developing content knowledge and teaching strategies. The third year is designed to refine authentic teaching skills through field practicums [7]. Throughout teacher education programs, OTVCs offer student teachers opportunities to observe, discuss, and reflect. In this regard, it is reasonable to expect that students at different levels may have different preferences and needs for these video resources. However, no study has investigated the impact of course year level on PPTs' perceived usefulness of OTVCs and their need for them in China.

## 2.2. *Five Domains Regarding Pre-Service Teachers' Video-Based Learning*

A substantial body of research is emerging that evaluates the advantages of various video-mediated training approaches for pre-service teachers in preschool, primary, or secondary education and explores how student teachers can benefit from such approaches. Within these studies [1,2], researchers have given particular attention to the following five domains.

### 2.2.1. Actors Showcased in the Videos

The first domain investigated in the present study is actors showcased in videos. Researchers have examined the effectiveness of videos featuring different actors in action, such as expert teachers, regular teachers, and participant teachers themselves [13,14]. Many videos feature expert teachers demonstrating effective or typical teaching strategies, classroom management skills, and engagement techniques. For instance, Lewis [14] conducted a study with elementary and secondary pre-service teachers with the facilitation of expert teacher videos. The results revealed that incorporating expert teacher videos in pre-service teachers' training programs could enhance their thinking on planning, instruction, and assessment. It was highly recommended that these master videos be used in future teacher field practice courses. Other studies have emphasized the use of videos showing participant teachers in pre-service teacher learning initiatives. These endeavors suggested that video material with a high level of personal relevance was more likely to stimulate in-depth thinking about teaching and learning. In the research conducted by McLeod [15], PPTs recorded videos of themselves and then engaged in self-reflection and received feedback from their peers based on these videos. The findings indicated that this video-mediated training format could be a highly effective method for implementing evidence-based practice with pre-service teachers.

### 2.2.2. Video-Mediated Learning Activities

The second domain considered herein is video-mediated learning activities. As evidenced in previous research, learning with videos is valuable in enhancing the pedagogical knowledge and skills of pre-service teachers. Diverse video-mediated activities have been found to be effective, particularly in the preschool field [16,17]. Pre-service teachers may benefit in multiple ways from individual learning with videos. Reflective journaling, for example, allows them to critically analyze their own teaching process through videos, leading to a deeper understanding of teaching and learning. Bayat [17] conducted action research using journaling and videos to promote productive reflection among PPTs during their field experience courses. The study demonstrated that individual video learning activities scaffolded student teachers' reflections, effectively connecting theory to practice. In addition, collaborative video learning with peers and instructors offers advantages for pre-service teachers. Collaborative learning activities with the support of video techniques expand their individual boundaries to achieve shared progress. Laparo, Maynard and other colleagues [16] designed a video review process to train PPTs in teacher–child interaction abilities. The study found that videotaping themselves and analyzing their strengths and challenges with peers and instructors helped pre-service teachers focus on specific behavioral objectives. Other video-mediated activities, such as recalling/describing events from videos or transcribing videos, can also yield positive results for PPTs. Cherrington [18] adopted video-stimulated recall interviews to help PPTs articulate their thinking and reflection on interactions with children. The results showed that this kind of recalling method was an effective attempt to assist PPTs in developing a shared understanding of teaching practice. These studies highlighted the usefulness of video-mediated learning activities, both individually and collaboratively, in enhancing the knowledge and skills of PPTs.

### 2.2.3. Learning Facilitators

The third domain focuses on the engagement or presence of facilitators while working with videos. Facilitators may play a crucial role in stimulating critical thinking by raising

questions and perspectives while observing the videos [19]. They also facilitate the collection and exchange of diverse viewpoints among teachers. For instance, Baecher and Jewkes [20] conducted a collaborative pre-service teacher preparation program for PPTs in conjunction with English language learning faculty. In this innovative video-based program, teacher educators from both preschool education and early language learning fields acted as facilitators, demonstrating to PPTs how to interpret and analyze instructions in videos. This collaborative learning paradigm has been demonstrated to support the instruction of early language teachers. Similarly, Mitchell and Marin [21] designed a video club to assist pre-service mathematics teachers in learning to notice. The role of the facilitator was to keep the group members on task and help them resolve any differences that arose during discussions. This learning framework was proven effective in helping pre-service teachers to notice more salient features of mathematics instruction.

### 2.2.4. Learning Situations

The fourth domain is the learning situations when videos are implemented. The researchers have recommended using video to boost pre-service teachers' learning across various situations [22,23]. The findings indicated that videos played a valuable role in supporting the learning of pre-service teachers in both generic and pedagogical skills coursework, as well as during their field-based practices. For instance, in a mathematics curriculum course, Beswick and Muir [22] used video excerpts to enhance the noticing ability of elementary pre-service teachers in mathematics. Findings showed that pre-service teachers expressed a positive attitude toward using videos in their courses. In another study, Kennedy and Lees [23] implemented video-based peer coaching and tiered support within a field-based practice module to develop appropriate adult–child interaction skills among PPTs. Results from the interviews and feedback indicated that PPTs experienced personal and professional growth during this module with the help of videos. These innovative approaches demonstrated that videos have been successfully integrated into diverse learning situations for pre-service teachers, effectively contributing to the preparation for their future careers.

### 2.2.5. Areas for Improvement

Finally, the fifth domain considered in this study was areas of improvement when learning with videos. Extensive research has provided compelling evidence that the use of classroom videos can significantly enhance various aspects of pre-service teachers' development, including instructional quality, content knowledge, teacher beliefs, and child-related outcomes [24,25]. For example, Garvis and Pendergast [25] conducted a study with PPTs, focusing on developing their professional skills in working with infants and toddlers. The findings emphasized the importance of utilizing videos as a tool to help pre-service teachers gain knowledge about children and develop their teacher identity through experiential learning. Similarly, McLeod and Kim [24] employed videos and email feedback to provide distance training for PPTs. Their study demonstrated the effectiveness of video-based instructions to improve their use in teaching strategies.

### 2.3. Research Questions

The domains reviewed in the prior section were identified in empirical studies conducted with different teacher populations. However, a significant research gap exists in understanding PPTs' perceptions of video and its alignment with their specific needs. Current arguments about classroom video are limited and lack a comprehensive understanding of the various perspectives on the impact of video. Specifically, there has been no research investigating PPTs' perceived usefulness of using OTVCs featuring different actors and posing specific OTVC-mediated activities. Similarly, we lack research on PPTs' needs with regard to facilitators, situations requiring OTVCs, and areas for improvement. Therefore, it is crucial to conduct additional research to address these gaps and explore their perceived usefulness and needs concerning OTVCs. This research will contribute to

a more holistic understanding of PPTs' perspectives on video-mediated learning and help teacher educators provide responsive resources and training opportunities.

We conducted this cross-sectional study to investigate the perceived usefulness of and need for OTVCs among PPTs in China and to compare potential differences among participants at different course year levels (Year 1, Year 2, and Year 3). The study was guided by the following two Research Questions (RQs):

RQ1: What are the levels of perceived usefulness of OTVCs among PPTs, specifically in the domain of the actors showcased in the OTVCs and in the domain of OTVC-mediated learning activities, and how do they differ across year levels?

RQ2: What are the needs related to OTVCs among PPTs, specifically in the domains of OTVC learning facilitators, situations requiring OTVCs, and areas for improvement, and how do the needs differ across year levels?

## 3. Method

### 3.1. Participants

This cross-sectional study targeted PPTs who were pursuing a three-year preschool teacher education program in five normal colleges in the Shandong province (China). A proportional stratified random sampling method was adopted to invite 300 PPTs from each of the three course year levels (Year 1, Year 2, and Year 3). The number of initial respondents was 805, and a total of 744 PPTs were finally recruited. All these PPTs were preschool education majors with similar academic backgrounds, receiving training to become preschool teachers for children aged 3-6 years. Most of them (77%) had more than one year of OTVC learning experience. The participants' ages ranged between 17 and 22 years (M = 20.12, SD = 1.09). Most of them (96.4%) were female (see Table 1). To achieve the goals of the study, we defined respondent groupings according to their year levels: 215 Year 1 participants (28.9%), 242 participants in Year 2 (32.5%), and 287 in Year 3 (38.6%).

**Table 1.** Demographics of pre-service participants (*n* = 744).

| Variable | Category | *n* | Percentage |
|---|---|---|---|
| Age | <20 years old | 442 | 59.4% |
| | >20 years old | 302 | 40.6% |
| Gender | Male | 27 | 3.6% |
| | Female | 717 | 96.4% |
| | Year 1 | 215 | 28.9% |
| Academic year | Year 2 | 242 | 32.5% |
| | Year 3 | 287 | 38.6% |
| | Less than 1 year | 171 | 23% |
| Years of watching | 1–2 years | 339 | 45.6% |
| OTVCs | 2–3 years | 173 | 23.3% |
| | More than 3 years | 61 | 8.1% |

### 3.2. Instruments

A survey, as detailed in the Appendix A, was designed specifically for this study. It collected data using a variety of response formats, including dichotomous, multiple-choice, and Likert scales. The survey consisted of three sections: (1) demographics: information about participants' gender, age, year level, and length of study with OTVCs; (2) perceived usefulness of OTVCs; (3) need for OTVCs.

To ensure the face, content, and ecological validity of our survey, we proceeded with a three-stage process as follows:

Stage 1: *Literature review and the initial draft survey design*. Following a comprehensive review of the literature on teachers' video-based learning, we developed a first draft of the survey, which included a series of exploratory questions and a preliminary list of response items related to the themes of perceived usefulness and need.

Stage 2: *Individual interviews for survey piloting*. We recruited four PPTs in each of the three year levels, a total of 12 PPTs, and conducted individual interviews with them. In these pilot interviews, participants were asked to think aloud and comment on the relevance and appropriateness of the survey items. They were also requested to suggest response options for survey questions on the extent of their knowledge of OTVCs. We made modifications according to their feedback to enhance the content validity and legibility as well as the flow of the survey.

Stage 3: *Final version of the survey and English-to-Chinese translation*. The modified version was then reviewed by two experts in preschool education and teacher education. As the survey was finally delivered in Chinese, an English-to-Chinese translation and back-translation were produced by two professional translators to ensure accuracy.

The final section "Perceived usefulness of OTVCs" contained the following items:

- *Actors showcased in the OTVCs*: "How useful would it be for you to watch OTVCs in which the actors are…?" Four options were listed, such as "Expert in-service teachers" and "Peers". Each item was followed by a 5-point Likert Scale ranging from Very Useless to Very Useful.
- *OTVC-mediated learning activities*: "Please indicate the extent to which the following activities are useful for your learning with OTVCs." Seven activities were listed, such as "Recalling/describing the events in videos" and "Collaborative reflection with peers upon videos, guided by instructors". Each item was followed by a 5-point Likert Scale ranging from Very Useless to Very Useful.

The final section pertaining to "Need for OTVCs" consisted of the following three items:

- *Need for learning facilitators*: "When learning with OTVC, to what extent do you need the following facilitators?" Three facilitators were presented, namely "Peers", "Instructors", and "Practitioners". Each facilitator was followed by a 5-point Likert Scale ranging from Not at All Needed to Very Much Needed.
- *Need for situations requiring OTVCs*: "To what extent do you need OTVCs in the following situations?" Five situations in which OTVCs may be needed were listed, such as "When preparing for teacher certification exams" and "When preparing for the practicum". Each situation was followed by a 5-point Likert Scale ranging from Not at All Needed to Very Much Needed.
- *Need for improvement*: "To what extent do you think you need OTVCs to improve in the following areas?" We presented six areas, such as "Confidence as a teacher" and "Classroom management skills". Each area was followed by a 5-point Likert Scale ranging from Not at All Needed to Very Much Needed.

The Cronbach's alpha values of the scale "Perceived usefulness of OTVCs" and the scale "Need for OTVCs" were 0.94 and 0.97, respectively, indicating a strong reliability of the whole survey.

### 3.3. Procedure

Ethical approval was first obtained from the Human Research Ethics Committee at the authors' university. The survey was released online via Tencent Survey, and was accessible for two weeks. We approached participants through WeChat and QQ (the most popular and typical social communication applications in mainland China). An invitation email was first sent to 12 students in these five normal colleges, which was then distributed to their classmates via WeChat or QQ groups. The email explained the overall purpose of our research and contained the information sheet and consent form as attachments, as well as the hyperlink to the online survey. Participants were notified that their participation was voluntary and anonymous, that they could quit the study at any time without negative consequences, and that they would not receive direct benefits or compensation. At the beginning of the survey, participants were asked to provide informed consent. The survey took them around 10 min to complete.

*3.4. Data Analysis*

A total of 805 participants initially responded to the survey for this study. Rigorous screening was conducted to ensure the quality of the responses. For example, responses that took less than 5 min to complete and respondents with no OTVC learning experience were eliminated. In the end, we deemed 744 responses valid for final data analysis. We first converted the response options for each item on the Likert scale into numerical values. To investigate RQ1, we generated individual tables for each of the two items of perceived usefulness. These tables presented the total mean score and standard deviation for each item, arranged in descending order. Additionally, the tables included the results for each year level, providing the rank position, mean score, and standard deviation for each item. The rankings, denoting the ordinal position (e.g., 1st, 2nd, 3rd), were determined based on the mean score of each respective group. Afterward, we performed analytic tests using one-way ANOVAs. The post hoc Tukey HSD or Games-Howell tests were also applied to detect whether there were differences between the three levels for each item. The same procedure was carried out to investigate RQ2 on OTVC needs. The SPSS for Mac version 27.0 was utilized for data analysis.

## 4. Results

### 4.1. Perceived Usefulness of OTVCs

As evidenced by the data presented in Table 2, participants generally attributed value to all categories of actors showcased in the OTVCs, with all total mean scores of the four items above 3 on a 1–5 scale. Participants in all three levels presented the most positive perceptions regarding in-service preschool teachers, including both "Expert in-service preschool teachers" (ranked first in all three groups) and "Regular in-service preschool teachers" (ranked second in all three groups). In comparison to in-service teachers, participants perceived OTVCs of their "Peer students" and "Own" as less useful, as indicated by their relatively lower scores.

**Table 2.** Perceived usefulness of actors showcased in the OTVCs.

| Category | Total Mean (SD) | Year 1 | | Year 2 | | Year 3 | |
|---|---|---|---|---|---|---|---|
| | | Rank | M (SD) | Rank | M (SD) | Rank | M (SD) |
| Expert in-service preschool teachers | 4.21 (0.91) | 1st | 4.14 (0.87) | 1st | 4.17 (0.90) | 1st | 4.30 (0.94) |
| Regular in-service preschool teachers | 3.99 (0.92) | 2nd | 3.78 (0.87) | 2nd | 3.91 (0.92) | 2nd | 4.21 * (0.91) |
| Your own | 3.63 (1.18) | 3rd | 3.33 (1.18) | 3rd | 3.65 * (1.13) | 3rd | 3.83 * (1.17) |
| Peer students | 3.62 (1.12) | 4th | 3.32 (1.06) | 4th | 3.60 * (1.13) | 4th | 3.85 * (1.11) |

Note: Asterisks [*] indicate the group with significant differences.

Table 2 also shows the items for which we detected significant differences between the three levels with respect to the perceived usefulness of the actors showcased. For the items, "Regular in-service preschool teachers", "Peer students", and "Your Own", one-way ANOVA tests indicated significant differences between the three levels ($F_{(2, 741)} = 15.437$, $p = 0.000$; $F_{(2, 741)} = 14.180$, $p = 0.000$; $F_{(2, 741)} = 11.511$, $p = 0.000$). For the items "Regular in-service preschool teachers" and "Your own", post hoc comparisons using the Tukey HSD test indicated that there were no significant differences between Year 1 and Year 2 participants ($p = 0.262$; $p = 0.154$). However, the mean scores of the Year 3 participants (M = 4.21, SD = 0.91; M = 3.84, SD = 1.17) were significantly higher than those of Year 1 and Year 2 ($p = 0.011$, $p = 0.00$; $p = 0.023$, $p = 0.00$). Post hoc comparisons were conducted with the Games-Howell test on the item "Peer students", which revealed that Year 3 participants (M = 3.85, SD = 1.11) reported a significantly higher mean than Year 1 and Year 2 (M = 3.32,

SD = 1.06, *p* = 0.000; M = 3.60, SD = 1.13, *p* = 0.032). The mean score of Year 2 was also significantly higher than Year 1 (*p* = 0.017).

Table 3 shows participants' perspectives on the usefulness of activities with OTVCs. All of the OTVC-mediated activities obtained scores higher than 3, indicating that participants regarded all activities as beneficial. They perceived collaborative activities with peers and instructors as the most useful. These activities included "Collaborative reflection with peers upon videos, guided by instructors" (ranked first in all three groups) and "Recalling/describing the events in videos" (ranked second in all three groups). In contrast, the individual learning activities and activities that required extensive time, such as "Individual reflection upon videos, guided by instructors", and "Transcribing videos", were placed lowest by participants.

**Table 3.** Perceived usefulness of OTVC-mediated activities.

| Category | Total Mean (SD) | Year 1 | | Year 2 | | Year 3 | |
|---|---|---|---|---|---|---|---|
| | | Rank | M (SD) | Rank | M (SD) | Rank | M (SD) |
| Collaborative reflection with peers upon videos, guided by instructors | 4.15 (0.96) | 1st | 4.07 (1.01) | 1st | 4.05 (0.96) | 1st | 4.30 * (0.90) |
| Recalling/describing the events in videos | 4.11 (0.97) | 2nd | 4.05 (0.99) | 2nd | 3.98 (0.99) | 2nd | 4.27 * (0.92) |
| Individual reflection upon videos, with no guidance | 4.03 (0.96) | 3rd | 3.88 (0.99) | 3rd | 3.97 (0.99) | 4th | 4.20 * (0.91) |
| Collaborative reflection with peers upon videos, with no guidance | 4.03 (0.97) | 4th | 3.87 (0.96) | 5th | 3.94 * (1.02) | 3rd | 4.23 * (0.91) |
| Coding/rating videos | 3.95 (1.02) | 5th | 3.71 (1.09) | 4th | 3.94 (0.99) | 6th | 4.13 * (0.96) |
| Individual reflection upon videos, guided by instructors | 3.92 (1.09) | 6th | 3.58 (1.16) | 6th | 3.91 (1.04) | 5th | 4.19 * (1.01) |
| Transcribing videos | 3.85 (1.12) | 7th | 3.44 (1.19) | 7th | 3.87 (1.06) | 7th | 4.13 * (1.01) |

Note: Asterisks [*] indicate the group with significant differences.

The data were analyzed to identify potential differences between the three levels, as evidenced in Table 3. One-way ANOVAs revealed that there were statistically significant differences between the three levels for all seven activities ($F_{(2, 741)}$ = 6.740, *p* = 0.001; $F_{(2, 741)}$ = 25.350, *p* = 0.000; $F_{(2, 741)}$ = 10.923, *p* = 0.000; $F_{(2, 741)}$ = 7.991, *p* = 0.000; ($F_{(2, 741)}$ = 20.600, *p* = 0.000; $F_{(2, 741)}$ = 10.720, *p* = 0.000; $F_{(2, 741)}$ = 5.754, *p* = 0.000). Post hoc Tukey HSD and Games-Howell tests were also conducted. Findings showed that Year 3 participants had significantly higher mean scores than Year 1 and Year 2 in six out of seven video-mediated activities: "Recalling/describing the events in videos"(*p* = 0.001), "Transcribing videos" (*p* = 0.000), "Individual reflection upon videos, with no guidance" (*p* = 0.000), "Individual reflection upon videos, guided by instructors" (*p* = 0.000), "Collaborative reflection with peers upon videos, with no guidance" (*p* = 0.000), "Collaborative reflection with peers upon videos, guided by instructors" (*p* = 0.003). However, when we conducted a post hoc Tukey HSD test on the item "Coding/rating videos", the results were different. The findings revealed that both Year 2 and Year 3 participants (M = 3.94, SD = 0.99, *p* = 0.000; M = 4.14, SD = 0.96, *p* = 0.000) scored higher than Year 1 participants (M = 3.71, SD = 1.09). There were no significant differences observed between Year 2 and Year 3 (*p* = 0.064).

### 4.2. Need for OTVCs

As for the need for facilitators, participants valued assistance from all three learning facilitators (see Table 4). However, they emphasized their demand for what preschool practitioners might provide when conducting video-based learning. "Practitioners (in-service teachers and principals)" ranked first for all three levels. On the other hand, the facilitators with less practical experience (i.e., "Peers") were ranked lowest.

**Table 4.** Need for learning facilitators.

| Category | Total Mean (SD) | Year 1 | | Year 2 | | Year 3 | |
|---|---|---|---|---|---|---|---|
| | | Rank | M (SD) | Rank | M (SD) | Rank | M (SD) |
| Practitioners (in-service teachers and principals) | 4.16 (0.96) | 1st | 4.14 (0.95) | 1st | 4.05 (1.01) | 1st | 4.28 * (0.91) |
| Instructors | 4.13 (0.95) | 2nd | 4.04 (1.01) | 2nd | 4.04 (0.98) | 2nd | 4.27 * (0.87) |
| Peers | 4.04 (1.00) | 3rd | 3.89 (1.04) | 3rd | 3.93 (1.01) | 3rd | 4.24 * (0.91) |

Note: Asterisks [*] indicate the group with significant differences.

One-way ANOVA analyses (F (2, 741) = 10.082, $p$ = 0.000; F (2, 741) = 5.469, $p$ = 0.004; F (2, 741) = 4.780, $p$ = 0.009) demonstrated significant differences in the needs for facilitators across the three levels (see Table 4). Regarding the item "Practitioners (in-service teachers and principals)", the post hoc Tukey HSD test only detected a significantly higher score in Year 3 participants (M = 4.29, SD = 0.91) when compared with Year 2 (M = 4.03, SD = 1.01, $p$ = 0.006). For the items "Peers" and "Instructors", post hoc Tukey HSD test results showed that no significant difference was discovered between Year 1 and Year 2 ($p$ = 0.916, $p$ = 0.999, respectively), but Year 3 participants (M = 4.24, SD = 0.91; M = 4.28, SD = 0.87) obtained significantly higher mean scores than the other two year levels on the item "Peers" (M = 3.89, SD = 1.04, $p$ = 0.001; M = 3.93, SD = 1.01, $p$ = 0.000), as well as the item "Instructors" (M = 4.04, SD = 0.98, $p$ = 0.015; M = 4.04, SD = 1.01, $p$ = 0.013).

Table 5 presents the findings pertaining to situations that may necessitate the use of OTVCs. Participants expressed a clear demand for OTVCs across all the mentioned situations, as indicated by a total mean score exceeding 4. Of all the situations offered, job preparation was the scenario when participants needed video the most for all levels. Survey participants listed "When preparing for teacher qualification exams" (ranked first or second) as the moment when they were in urgent need of OTVCs. For the item "When preparing for practicum", responses varied across the three levels. We found that Year 1 and Year 3 participants placed practicum as a top priority (ranked second and first, respectively), while Year 2 ones placed it lower. We also found that there was an urgent need for Year 2 participants "When preparing for course exams", whereas this situation was only middle-ranked and bottom-ranked by Year 1 and Year 3 responders. Furthermore, participants assigned low to medium ratings for the need for OTVCs when it came to "When doing self-directed learning" and "When taking compulsory courses".

**Table 5.** Need for situations requiring OTVCs.

| Category | Total Mean (SD) | Year 1 | | Year 2 | | Year 3 | |
|---|---|---|---|---|---|---|---|
| | | Rank | M (SD) | Rank | M (SD) | Rank | M (SD) |
| When preparing for teacher certification exams | 4.25 (0.93) | 1st | 4.26 (0.97) | 1st | 4.16 (0.92) | 2nd | 4.31 (0.90) |
| When preparing for the practicum | 4.21 (0.92) | 2nd | 4.18 (0.97) | 4th | 4.08 (0.96) | 1st | 4.33 * (0.84) |
| When preparing for course exams | 4.15 (0.93) | 3rd | 4.16 (1.00) | 2nd | 4.11 (0.92) | 5th | 4.26 (0.89) |
| When doing self-directed learning | 4.14 (0.93) | 5th | 3.99 (1.02) | 3rd | 4.11 (0.93) | 3rd | 4.28 * (0.85) |
| When taking compulsory courses | 4.13 (0.92) | 4th | 4.00 (0.92) | 5th | 4.07 (0.94) | 4th | 4.27 * (0.88) |

Note: Asterisks [*] indicate the group with significant differences.

Table 5 presents the significant differences identified through the one-way ANOVA tests for the situation needs between the three levels. For the items "When taking compulsory courses", "When doing self-directed learning", and "When preparing for practicum",

one-way ANOVA tests indicated significant differences between the three levels (F (2, 741) = 6.504, *p* = 0.002; F (2, 741) = 6.310, *p* = 0.002; F (2, 741) = 4.965, *p* = 0.007). For the item "When taking compulsory courses", post hoc comparisons using the Tukey HSD test indicated that there were no significant differences between Year 1 and Year 2 participants (*p* = 0.655). However, the mean scores of the Year 3 participants (M = 4.28, SD = 0.88) were significantly higher than those of Year 1 and Year 2 (M = 4.00, SD = 0.92, *p* = 0.002; M = 4.07, SD = 0.94, *p* = 0.027). By a post hoc Tukey HSD test on the item "When doing self-directed learning", we only found that Year 3 participants (M = 4.28, SD = 0.85) had significantly higher means than Year 1 participants (M = 3.99, SD = 1.02, *p* = 0.002), while there were no significant differences between Year 1 and Year 2 (*p* = 0.372), nor between Year 2 and Year 3 (*p* = 0.079). As for the item "When preparing for practicum", the post hoc Tukey HSD test only detected a significantly higher score of Year 3 participants (M = 4.33, SD = 0.84) when compared with Year 2 (M = 4.08, SD = 0.96, *p* = 0.006).

As shown in Table 6, all six categories regarding needs for improvement scored above 4, indicating that participants had strong needs in these domains when learning with OTVCs. Our data revealed similarities and differences in teachers' needs for improvement. Regarding the highest-priority needs, Year 1 and Year 3 participants ranked "Content knowledge (i.e., knowledge about health, language, society, science, and art)" the highest. Their other needs, in descending order, were "Able to achieve better learning outcomes in children" and "Instructional quality". Compared to the other two levels, Year 2 participants had a completely different set of preferred needs for improvement. Their strongest need was "Able to achieve better learning outcomes in children", followed by "Classroom management skills" and "Content knowledge". Improvement areas such as "Confidence as a teacher" and "Identity as a teacher" were ranked lowest across all three levels.

**Table 6.** Need for improvement.

| Category | Total Mean (SD) | Year 1 | | Year 2 | | Year 3 | |
|---|---|---|---|---|---|---|---|
| | | Rank | M (SD) | Rank | M (SD) | Rank | M (SD) |
| Content knowledge (health, language, society, science, and art) | 4.18 (0.92) | 1st | 4.03 (1.00) | 3rd | 4.11 (0.93) | 1st | 4.35 * (0.82) |
| Able to achieve better learning outcomes in children | 4.16 (0.92) | 2nd | 4.00 (0.99) | 1st | 4.12 (0.92) | 3rd | 4.32 * (0.85) |
| Instructional quality | 4.16 (0.93) | 3rd | 4.00 (1.00) | 4th | 4.10 (0.94) | 2nd | 4.32 * (0.84) |
| Classroom management skills | 4.15 (0.91) | 4th | 3.99 (0.98) | 2nd | 4.11 (0.92) | 4th | 4.31 * (0.82) |
| Confidence as a teacher | 4.13 (0.95) | 5th | 3.99 (1.01) | 6th | 4.07 (0.94) | 5th | 4.28 * (0.90) |
| Identity as a teacher | 4.10 (0.96) | 6th | 3.93 (1.00) | 5th | 4.08 (0.95) | 6th | 4.26 * (0.92) |

Note: Asterisks [*] indicate the group with significant differences.

One-way ANOVA tests showed that significant differences were found between the three levels for all six areas (F (2, 741) = 6.740, *p* = 0.001; F (2, 741) = 8.283, *p* = 0.000; F (2, 741) = 9.677, *p* = 0.000; F (2, 741) = 8.649, *p* = 0.000; (F (2, 741) = 7.578, *p* = 0.001; F (2, 741) = 6.304, *p* = 0.002; F (2, 741) = 7.985, *p* = 0.000). Post hoc Tukey HSD tests were then conducted. The findings showed that while there were no significant differences between Year 1 and Year 2, Year 3 participants had significantly higher mean scores than the other two levels in five of the six areas: "Instructional quality" (*p* = 0.001), "Classroom management skills" (*p* = 0.000), "Content knowledge" (*p*=.000), "Confidence as a teacher" (*p* = 0.002), and being "Able to achieve better learning outcomes in children" (*p* = 0.000). The result of a post hoc Tukey HSD test on the item "Identity as a teacher" was different. The findings revealed that only Year 3 participants (M = 4.26, SD = 0.92, *p* = 0.000) had a significantly higher score than Year 1 participants (M = 3.93, SD = 1.00). However, no sig-

nificant differences were observed between Year 2 and Year 3 ($p = 0.089$) or between Year 1 and Year 2 ($p = 0.184$).

## 5. Discussion

RQ1 aimed at investigating the perceived usefulness of OTVCs among PPTs, with a focus on the actors showcased in the OTVCs and the OTVC-mediated learning activities, comparing across the three levels of preschool teacher education programs in China. Regarding the actors showcased in OTVCs, participants generally viewed all types of actors (i.e., expert in-service teachers, regular in-service teachers, and their own) as valuable. There were significant differences between the three levels, with Year 3 PPTs considering the actors presented as significantly more useful than lower-level PPTs. Notably, participants reported a strong preference for expert teacher demonstrations. They ranked expert in-service teachers as the top priority. While previous scholars have emphasized the importance of teachers viewing themselves [1,15], our findings support the notion that PPTs prefer to see seasoned educators in the OTVCs and perceive expert modeling as more useful for triggering their pedagogical minds, which aligns with Lewis's [14] findings on the usefulness of expert videos integrated into pre-service teachers' preparation courses.

In terms of the OTVC-mediated activities, the participants perceived all the activities listed in the survey as useful, with Year 3 respondents finding them particularly valuable. Although significant differences were detected, participants, especially the Year 3 PPTs, expressed a clear preference for engaging in discussions with peers and receiving instructor feedback over other OTVC-mediated activities, such as individual learning activities (i.e., self-reflection with and without guidance). The recognition of the importance of direct interaction with peers and instructors echoes the findings of Kennedy and Lees [23], supporting the proposition that PPTs prefer conducting video observation activities collectively under guidance. Other preschool scholars, such as McLeod and Kim [24] and Baecher and Jewkes [20], have suggested that video-mediated activities that combine individual and collaborative learning can be a more viable support for PPTs. Furthermore, it is noteworthy that recalling and describing video activity ranked second in our study. Year 3 PPTs found this activity more beneficial compared to other levels. This finding is consistent with the study of Cherrington [18], which used recall tasks to train PPTs. Similarly, Bautista, Ho, and colleagues [2] also called for more investigation into the effectiveness of recalling and describing video activities. Our results reflected the perspectives of PPTs who viewed the detailed recall of OTVCs as valuable.

The purpose of RQ2 was to investigate the need of PPTs for OTVCs, specifically in the domains of learning facilitators, situations requiring OTVCs, and areas for improvement, comparing the needs of PPTs across the three levels. Regarding facilitators, the respondents reported a critical need for in-service preschool teachers to serve as their OTVC learning facilitators, although there were significant differences across year levels, with Year 3 PPTs expressing a more urgent need. This finding echoes the prior study conducted by Kurz, Batarelo, and Middleton [6], reflecting the desire of PPTs to observe experienced experts in order to help them visualize key aspects of the teaching profession. The participants also expressed the need to engage with larger communities of preschool practitioners for future career preparation [7].

Regarding the need for OTVCs in various situations, participants from different year levels indicated that job preparation was their primary concern, with Year 3 participants being most concerned about this issue. Course learning is the situation where most of the available literature has recommended using videos among pre-service teachers [22,23]. However, when we investigated this domain through the lens of PPTs, our result was novel and worth noting. Our participants ranked preparing for the qualification exam as the primary situation in which they needed OTVCs. This finding coincides with the current social realities and expectations in China. In recent decades, China has launched several policies stating that it is necessary to popularize high-quality preschool education and to raise the requirements for the professional competencies of preschool teachers [26,27]. The rapid de-

velopment of preschool education in China and the increasing demands on teachers may explain the urgent need for PPTs to use OTVCs in their job preparation to gain insight into the challenges and dynamics of teaching preschool children.

In terms of areas for improvement, despite the differences across year levels, participants reported a clear need to improve their content knowledge in the preschool field. Content knowledge in five learning fields (health, language, society, science, and art) ranked highest, suggesting that PPTs have an explicit need for OTVCs to stock up on the knowledge to teach specific learning areas and the corresponding pedagogical approaches. In China, pre-service teacher education programs take an integrated, holistic approach to preparing PPTs [27]. This broad curriculum may lack guidance for teachers on content knowledge, leaving PPTs in need of additional assistance from OTVCs to support children's learning in specific areas.

This study has made a remarkable contribution by uncovering significant differences in PPTs' perceived usefulness of and need for OTVCs between different year levels. Our results revealed that Year 3 respondents reported the highest mean scores in almost all items investigated. This may be partly due to the fact that they have received more formal training with OTVCs and have had relatively more opportunities to engage in teaching practice. As graduation approaches, Year 3 PPTs may have a greater urgency to utilize OTVCs and enhance their pedagogical skills. This finding is consistent with the trends reported by Ambrosetti [11], reflecting a greater willingness of final-year student teachers to improve themselves and refine their teaching abilities.

## 6. Conclusions

We innovatively developed a quantitative survey to measure the PPTs' perceptions of video-based learning. Based on our findings, we conclude that Chinese PPTs varied in their perceptions of the usefulness of and need for OTVCs. In particular, they expressed relatively high levels of usefulness and need for the following OTVCs in the five domains considered in the study:

1.  OTVCs featuring expert teachers who illustrate exemplary or typical instructional practices.
2.  OTVC-mediated activities that promote collaboration and interaction and allow for the recalling of teaching events.
3.  OTVC learning facilitated by expert in-service preschool teachers for hands-on guidance.
4.  OTVCs that assist in job preparation for future teaching roles.
5.  OTVCs which cover subject matter content knowledge in five learning fields.

Additionally, we found that the course year level is a crucial variable in determining PPTs' perceived usefulness of and need for OTVCs, which should be considered when designing PPT training programs. Different OTVC strategies should be adopted to address the specific needs of PPTs at different stages of their training.

### 6.1. Limitations and Future Research

The present study has various limitations that must be acknowledged. First, while our methodology enabled us to survey a relatively large number of PPTs, the study relied exclusively on only one data source. In future studies, other data sources (e.g., interviews) and more supplemental data should be used for triangulation to gain a deeper understanding of PPTs' perceived usefulness of and need for OTVCs. Second, this study only focused on investigating PPTs in one region of China. The results may not be generalizable to other regions or other countries. Conducting similar research with other samples of PPTs would provide further support and validation for our findings. Finally, this study solely aimed to understand PPTs' perceived usefulness of and need for OTVCs during their pre-service education period. It will be of great interest to explore whether pre-service teachers' preferences and needs change as they gain teaching experience. Conducting longitudinal

studies to track changes in PPTs throughout their teaching careers would provide valuable insights into the evolving nature of their preferences and need for OTVCs.

*6.2. Implications*

Our study offers valuable insights into the design of OTVC-mediated training initiatives for PPTs. First, we found that PPTs perceived OTVCs as highly useful and expressed a clear need to incorporate OTVCs into their learning process. Based on these findings, we offer the following suggestions for teacher educators to make OTVC-mediated instruction responsive to PPTs' preferences and needs:

1. Utilize expert videos that demonstrate exemplary teaching practices as video resources.
2. Prioritize collaborative activities that engage peers and instructors, as well as activities that allow PPTs to recall teaching events through videos.
3. Consider selecting facilitators who are preschool practitioners with practical experience and capable of offering expert advice, guidance, and demonstration.
4. Prepare OTVCs that meet PPTs' needs for job readiness.
5. Provide precise OTVC materials to help PPTs with subject matter content knowledge.

Teacher educators should also consider the significant differences in the perceptions and demands between PPTs at different course year levels. New-entry PPTs may need comprehensive support in all areas to build their self-efficacy, while final-year PPTs may benefit from more precise aids to prepare them for their future career. Therefore, it is crucial to provide OTVC materials with different emphases to meet their different needs.

Additionally, given the varying policy priorities in different regions and the rapidly changing demand for preschool teachers, it is essential to provide PPTs with responsive teacher education [28,29]. Teacher educators should prioritize the perceptions and needs of PPTs and develop training programs that not only reflect local policy priorities but also align with their preferences. It is important to continuously investigate the perceived usefulness of and need for PPTs, given that these perceptions and needs are contextualized and may evolve over time. Further exploration of this topic should be conducted with different teacher populations to ensure an updated understanding of PPTs' perceptions and needs.

**Author Contributions:** Conceptualization, R.X. and A.B.; methodology, R.X.; software, R.X.; formal analysis, R.X.; data curation, R.X.; writing—review and editing, R.X., A.B. and W.Y.; supervision, A.B. All authors have read and agreed to the published version of the manuscript.

**Funding:** This research received no external funding.

**Institutional Review Board Statement:** The study was conducted in accordance with the Declaration of Helsinki and approved by the Human Research Ethics Committee of the Education University of Hong Kong (Ref. no. 2022-2023-0407; 18 July 2023).

**Informed Consent Statement:** Informed consent was obtained from all subjects involved in the study.

**Data Availability Statement:** The data presented in this study are available on request from the corresponding author due to privacy and ethical restrictions.

**Conflicts of Interest:** The authors declare no conflicts of interest.

## Appendix A

Survey on the Perceptions of Perceived Usefulness and Need for Online Teaching Video Cases (OTVCs)
有关在线教学视频案例有效实践和未来需求的调查研究

(1) Demographics

Age 年龄
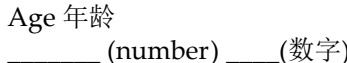 (number) ____(数字)

Gender 性别

Male 男

Female 女

Prefer not to say 不方便透露

Which year within your program are you currently enrolled in?

你目前处于哪个年级?

Year 1 一年级

Year 2 二年级

Year 3 三年级

Have you watched any OTVCs?

你是否看过在线教学视频教学案例?

No, not at all (end of the survey) 完全没有

"If you choose this option, it means that you haven't watched any OTVCs provided by instructors in any compulsory courses or searched by yourself online." 如果您选择此选项,则表示您没有观看过任何必修课程中由教师提供的或自己在网上搜索的在线视频课例。

Yes, very few 有一点

Yes, some 有一些

Yes, a lot 有很多

How many years have you used OTVCs during your program?

你有几年观看在线教学视频案例的经历?

Less than 1 year 少于一年

1–2 years 一至两年

2–3 years 二至三年

3–4 years 三至四年

(2) Perceived usefulness of OTVCs

**Table A1.** How useful would it be for you to watch OTVCs in which the actors are…您认为观看以下展示者的在线教学视频课例有多大帮助?

| | Very Useless | Useless | Neutral | Useful | Very Useful |
|---|---|---|---|---|---|
| Expert in-service preschool teachers 专家幼儿园教师 | | | | | |
| Regular in-service preschool teachers 普通幼儿园教师 | | | | | |
| Peer students 同学 | | | | | |
| Your own 自己 | | | | | |

**Table A2.** Please indicate the extent to which the following activities are useful for your learning with OTVCs. 请指出下列活动在多大程度上有助于您使用在线教学视频课例进行学习。

| | Very Useless | Useless | Neutral | Useful | Very Useful |
|---|---|---|---|---|---|
| Recalling/describing the events in videos 回顾/描述视频中的事件 | | | | | |
| Transcribing videos 转录视频 | | | | | |
| Coding/rating videos 对视频进行编码注释/评级 | | | | | |
| Individual reflection upon videos, with no guidance 独立反思 | | | | | |

**Table A2.** *Cont.*

|  | **Very Useless** | **Useless** | **Neutral** | **Useful** | **Very Useful** |
|---|---|---|---|---|---|
| Individual reflection upon videos, guided by instructors<br>在教师的指导下独立反思 |  |  |  |  |  |
| Collaborative reflection upon videos with peers, with no guidance<br>与同学独自进行合作反思 |  |  |  |  |  |
| Collaborative reflection upon videos with peers, guided by instructors<br>在教师的指导下与同学合作反思 |  |  |  |  |  |

(3) Need for OTVCs

**Table A3.** When learning with OTVC, to what extent do you need the following facilitators? 当使用在线教学视频课例学习时,您在多大程度上需要以下促进者?

|  | **Not at All Needed** | **Not Very Needed** | **Undecided** | **Somewhat Needed** | **Very Much Needed** |
|---|---|---|---|---|---|
| Peers<br>与同学一起学习 |  |  |  |  |  |
| Instructors<br>与老师一起学习 |  |  |  |  |  |
| Practitioners (in-service teachers and principals)<br>与幼儿园教师和园长一起学习 |  |  |  |  |  |

**Table A4.** To what extent do you need OTVCs in the following situations? 在以下情况中,您在多大程度上需要在线教学视频课例?

|  | **Not at All Needed** | **Not Very Needed** | **Undecided** | **Somewhat Needed** | **Very Much Needed** |
|---|---|---|---|---|---|
| When taking compulsory courses<br>必修课程 |  |  |  |  |  |
| When doing self-directed learning<br>自我学习 |  |  |  |  |  |
| When preparing for course exams<br>准备课程考试 |  |  |  |  |  |
| When preparing for the practicum<br>准备参加实习 |  |  |  |  |  |
| When preparing for teacher certification exams<br>准备教资考试 |  |  |  |  |  |

**Table A5.** To what extent do you think you need OTVCs to improve in the following areas? 您认为需要在线教学视频课例在多大程度上可以改进以下方面的工作?

|  | **Not at All Needed** | **Not Very Needed** | **Undecided** | **Somewhat Needed** | **Very Much Needed** |
|---|---|---|---|---|---|
| Instructional quality<br>教学能力 |  |  |  |  |  |
| Classroom management skills<br>课堂管理技巧 |  |  |  |  |  |

**Table A5.** *Cont.*

| | Not at All Needed | Not Very Needed | Undecided | Somewhat Needed | Very Much Needed |
|---|---|---|---|---|---|
| Content knowledge (health, language, society, science, and art) 学科内容 | | | | | |
| Identity as a teacher 教师认同感 | | | | | |
| Confidence as a teacher 成为教师的信心 | | | | | |
| Able to achieve better learning outcomes in children 能够实现儿童更好的学习收获 | | | | | |

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
