# Peer review of "Learning with Online Teaching Video Cases: Investigating Pre-Service Preschool Teachers’ Perceived Usefulness and Needs"

_education, doi:10.3390/educsci14050479_

Round 1
Reviewer 1 Report
Comments and Suggestions for Authors
There is excellent work for scholars and practitioners.
I have a question and a suggestion about the RQ. On p. 2, you discussed five domains regarding pre-service teachers´ video-based learning. But when you formulate the two RQs, you choose part of them, not all. Why? Why part of the domains and just these domains and not others?
You can do it, but you must explain your considerations.
The other question concerns the participants (3.1). I see the explanations in the research's limitations, but that is insufficient. You must justify the sample in Methods (p. 3).
Author Response
- I have a question and a suggestion about the RQ. On p. 2, you discussed five domains regarding pre-service teachers´ video-based learning. But when you formulate the two RQs, you choose part of them, not all. Why? Why part of the domains and just these domains and not others? You can do it, but you must explain your considerations. Response: Thanks for your valuable comments. We have added several sentences to explain why we chose two domains for perceived usefulness and the others for needs. (Line 186-190)
- The other question concerns the participants (3.1). I see the explanations in the research's limitations, but that is insufficient. You must justify the sample in Methods (p. 3). Response: Thank you so much for your suggestions. The sampling method has been added to the manuscript. (Line208-211)
Reviewer 2 Report
Comments and Suggestions for Authors
This paper is well written and documents the research clearly and comprehensively. The conclusions are sound.
This paper is acceptable in its present form though the following suggestions are to improve clarity for an international audience:
1. indicate the age range of children these preschool teachers (PPTs) would be teaching.
2. Line e.g.s 16, 75, 190, 206, 421 – the use of the term ‘grade’ is confusing when talking about the years of a teacher education course. Elsewhere the authors refer to year level. Maybe say ‘years of a teacher education course’ or ‘course year level’
3. Line 184-187 These sentences should be redrafted to be clearer and more understated in claims of significance.
Author Response
- indicate the age range of children these preschool teachers (PPTs) would be teaching. Response: Thank you so much for your comments. We have added this information. (Line 212-213)
- Line e.g.s 16, 75, 190, 206, 421 – the use of the term ‘grade’ is confusing when talking about the years of a teacher education course. Elsewhere the authors refer to year level. Maybe say ‘years of a teacher education course’ or ‘course year level’ Response: To tackle this comment, we have modified all the "grade" into "course year level" or "year level". Thanks!
- Line 184-187 These sentences should be redrafted to be clearer and more understated in claims of significance. Response: Done, thanks!